# Breast Cancer Tumor Stroma: Cellular Components, Phenotypic Heterogeneity, Intercellular Communication, Prognostic Implications and Therapeutic Opportunities

**DOI:** 10.3390/cancers11050664

**Published:** 2019-05-13

**Authors:** Noemi Eiro, Luis O. Gonzalez, María Fraile, Sandra Cid, Jose Schneider, Francisco J. Vizoso

**Affiliations:** 1Research Unit, Fundación Hospital de Jove, Avda. Eduardo Castro, 161, 33290 Gijón, Spain; investigacion@hospitaldejove.com (M.F. & S.C.); 2Department of Anatomical Pathology, Fundación Hospital de Jove, Avda. Eduardo Castro, 161, 33290 Gijón, Spain; a.patologica2@hospitaldejove.com; 3Department of Obstetrics and Gynecology, Universidad Rey Juan Carlos, Avda. de Atenas s/n, 28922 Alcorcón, Madrid, Spain; jose.schneider@urjc.es; 4Department of Surgery, Fundación Hospital de Jove, Avda. Eduardo Castro, 161, 33290 Gijón, Spain

**Keywords:** CAFs, MMPs, TIMPs, cytokines, TLRs, mesenchymal stromal cells, exosomes, integrins

## Abstract

Although the mechanisms underlying the genesis and progression of breast cancer are better understood than ever, it is still the most frequent malignant tumor in women and one of the leading causes of cancer death. Therefore, we need to establish new approaches that lead us to better understand the prognosis of this heterogeneous systemic disease and to propose new therapeutic strategies. Cancer is not only a malignant transformation of the epithelial cells merely based on their autonomous or acquired proliferative capacity. Today, data support the concept of cancer as an ecosystem based on a cellular sociology, with diverse components and complex interactions between them. Among the different cell types that make up the stroma, which have a relevant role in the dynamics of tumor/stromal cell interactions, the main ones are cancer associated fibroblasts, endothelial cells, immune cells and mesenchymal stromal cells. Several factors expressed by the stroma of breast carcinomas are associated with the development of metastasis, such as matrix metalloproteases, their tissular inhibitors or some of their regulators like integrins, cytokines or toll-like receptors. Based on the expression of these factors, two types of breast cancer stroma can be proposed with significantly different influence on the prognosis of patients. In addition, there is evidence about the existence of bi-directional signals between cancer cells and tumor stroma cells with prognostic implications, suggesting new therapeutic strategies in breast cancer.

## 1. Introduction

Breast cancer is the most frequent malignant tumor in women and the leading cause of cancer death, since 30% of breast cancers develop distant metastases after the initial treatment of the apparently localized tumors [1]. Nowadays, the mechanisms underlying the genesis and progression of breast cancer are better understood [2], but despite an improvement of the survival rates for some molecular subtypes of breast cancer, we are still far from a cure for all patients [3]. Furthermore, classical (chemotherapy and radiation therapy) or new therapeutic strategies (selective targeting of oncogenes, immune toxicity or oncolytic virotherapy), are far from satisfactory and often associated with significant side effects, including collateral damage, drug resistance, immune toxicity or virus mutability and unexpected toxicity. It seems increasingly clear that the old dogma of cancer based only on a malignant transformation of the epithelial cells is too simplistic, and a new concept considering cancer as an ecosystem based on a cell sociology and the tumor-stroma crosstalk, is gaining strength. A better knowledge of the role of tumor stroma and its interaction with cancer cells can lead us to implement new prognostic tools or new therapeutic strategies aiming at a disruption of the dynamics of tumor-stroma interactions.

In the present work, we reviewed several key pathophysiological aspects related to tumor stroma and breast cancer progression, their clinical implications and possible therapeutic opportunities.

## 2. Tumor Stroma Components

The tumor stroma consists of the non-malignant cells of the tumor mass. Among the various cell types that make up the tumor stroma, and play a key role in tumor-stroma interactions, we mainly find resident cells such as cancer-associated fibroblasts (CAFs), endothelial cells and perycites, blood derived cells such as immune cells, and mesenchymal stroma cells which may be resident or attracted by the tumor itself and sometimes, by adipocytes surrounding it [1].

### 2.1. Cancer-Associated Fibroblasts (CAFs)

Cancer-asssociated-fibroblasts (CAFs) are one of the most abundant cell types in breast cancer stroma with a well recognized role in the initiation and progression of tumor progression. The CAF population derives from resident fibroblasts, but CAFs can also stem from other origins, including mesenchymal stromal cells (MSCs), epithelial cells, pericytes, adipocytes and endothelial cells [2]. CAFs differ from normal fibroblasts by showing a different gene expression profile and promoting cancer cell aggressiveness [3,4,5].

However, although it has been proposed that contractile forces exerted by CAFs can alter the organization and the physical properties of the basement membrane (interface of epithelium and stroma), making it permissive for cancer cell invasion [6], the role of CAFs in tumor progression is more complex. CAFs show a high proliferation rate and can induce the degradation and remodeling of the extracellular matrix (ECM), epithelial mesenchymal transition (EMT) activation, angiogenic shift, metabolic reprogramming toward a reverse Warburg phenotype, or promote stem cell trait achievement, as compared with normal fibroblasts [7,8,9]. The influence of CAFs is effected through a paracrine function by means of the secretion of growth factors and cytokines [10,11,12,13], such as IL-1β, IL-6, IL-8, SDF-1, and NFκB, in order to induce immune cell recruitment that may contribute to tumor progression [14,15]. CAFs are not only able to promote cancer progression but also cancer survival, by means of the creation of a “protective niche” that maintains residual tumor cell survival, such as through the induction of resistance to cancer therapy. In this sense, secretion of hepatocyte growth factor (HGF) and IL-6 by CAFs has been related to lapatinib resistance in HER2+ breast cancer [16] and tamoxifen resistance [17], respectively.

### 2.2. Immune Cells

The immune system plays a complex role in tumorigenesis [18] and immune cells, along with CAFs, are one of the main cell populations making up the tumor mass in invasive breast carcinomas. Tumor-infiltrating leukocytes have been historically considered to be manifestations of an intrinsic defense mechanism against developing tumors [19] and also, subsequently, interpreted as an aborted attempt of the immune system to reject the tumor. However, nowadays, it is well known that leukocyte infiltration can promote tumor growth, angiogenesis and tumor cell invasion [20,21], due to the secretion of cytokines, growth factors, chemokines and proteases [22,23,24].

The immune cell infiltrate includes a variable representation of leukocytes, including macrophages, neutrophils, mast cells, T- and B-lymphocytes [20]. Breast carcinomas may have different types of immune cell infiltrate able to control tumor growth. In this sense, the infiltration of macrophages, also named tumor-associated macrophages (TAMs) and known to have pro-tumoral functions, has been associated with a worse prognosis [20,25,26]. Macrophages can be polarized into two phenotypes: classically activated (M1) macrophages and alternatively activated (M2) macrophages driven by a cytokine repertoire of T helper cells (Th1 or Th2). M1 has been established as a tumor-suppressive phenotype and M2 as a tumor-promoting phenotype, considering that TAMs are primarily M2 polarized. On the other hand, the presence of both T- and B-cells in the microenvironment has been related to an immunological response, inhibiting cancer development and progression [27,28,29,30,31,32,33,34,35]. In this sense, a subgroup of CD4^+^ T helper cells, known as regulatory T cells (Tregs), has been associated with the suppression of T-cell immunity. Forkhead box protein P3 (FoxP3) is a transcription factor able to induce the immunosuppressive function of Tregs, being the most specific marker for Tregs in tumors. Due to the ability to inhibit anti-tumor immunity, tumor-infiltrating Foxp3+ Tregs were associated with poor prognosis [36], but recently, it has been shown that Foxp3+Tregs could improve survival in some tumors [37,38]. In a previous study, we described the clinical relevance of the relative amount of macrophages (CD68^+^), T-cells (CD3^+^) and B-cells (CD20^+^) at the invasive front of breast carcinomas. Thus, an increased CD68 count and CD68 / (CD3 + CD20) ratio were both directly associated with a higher probability of shortened relapse-free survival [39]. Nevertheless, the prognostic significance of the immune cell infiltrate in the tumor microenvironment remains controversial, perhaps due to non-standardized evaluation. Therefore, it is important, and possible as described below, to identify new factors as markers of immune stromal cells associated with tumor aggressiveness. Due to the important role of the host immune system in cancer, immune checkpoint inhibitors have garnered attention during the last years, especially against cytotoxic T-lymphocyte antigen-4 (CTLA-4) and programmed cell death protein 1 (PD-1) or its ligands (PDL-1) [40].

### 2.3. Endothelial Cells

Endothelial cells (ECs) are ubiquitous within tumors and necessary for the development and functionality of vessels, especially blood vessels, essential to supply oxygen and nutrients for tumor progression. The endothelial barrier maintains vascular and tissue homeostasis but its alteration leads to vascular permeability and drives tumor-induced angiogenesis, blood flow disturbances, inflammatory cell infiltration and tumor cell extravasation. In addition, regardless of perfusion, ECs can regulate tumor growth through the secretion of paracrine factors, thereby increasing tumorigenicity, stemness and invasiveness [41,42,43,44]. All of these effects are due to the crosstalk between tumor and ECs, leading to the initiation of angiogenesis [45] and also to the development of an abnormal phenotype of ECs, which can be different depending on tumors. Thus, for example, it has been reported that ECs from highly metastatic tumors have a more proangiogenic phenotype than those from low metastatic tumors [46].

### 2.4. Mesenchymal Stromal Cells

Mesenchymal stromal cells (MSCs) are adult multipotent stromal cells, which exhibit certain common properties, ranging from the expression of surface markers (CD73, CD90, CD105), to self-renewal capability and differentiation into osteoblasts, chondrocytes and adipocytes [47,48]. MSCs preferentially reside in perivascular niches of nearly all neonatal human tissues and organs. MSCs are recruited to sites of injury where their functions are extremely diverse and depend on the tissue-specific origins and the microenvironment in which MSCs are embedded. Migration of MSCs towards the inflammation site leads to cellular interactions that occur both directly via gap junctions, membrane receptors and nanotubes and indirectly via soluble structures and factors. In physiological situations, MSCs contribute to support tissue repair, stem cell homeostasis and immunomodulation. MSCs interact with the surrounding cells by means of the secretion of soluble factors, such as cytokines or growth factors, with activities that can dramatically alter the key cellular functions of neighboring cells, such as survival, apoptosis, maturation and differentiation [49].

However, MSCs and their paracrine-based activity also have a relevant role in the tumor-stroma crosstalk [50,51], since MSCs themselves can be stimulated by tumor cells to develop an aberrant tumor-associated phenotype [52]. Tumor cells can recruit MSCs to the tumor site using chemotactic factors such as MMPs, growth factors and inflammatory cytokines [53]. In turn, MSCs can release paracrine signals, including cytokines, chemokines and growth factors [51], to stimulate neighboring cells with pro- and/or anti-tumorigenic activities. Thereby, MSCs, having the capability to migrate to the tumor site [54,55], display several pro-tumor functions, such as the promotion of tumor growth [56] and angiogenesis [57,58,59], epithelial-to-mesenchymal transition (EMT) induction [60,61], as well as alteration of the extracellular matrix [62,63] to promote the migration and implantation of tumor cells to metastatic sites [54,64]. All of these effects, in their turn, are attained mainly by means of the secretion of pro-inflammatory cytokines [65], the suppression of immune effector cells [66,67,68,69,70,71], the expansion of immune regulatory cells [66,70,72] and increasing resistance to cancer therapies [73,74].

Additionally, although as previously mentioned, MSCs may be a source of CAFs, MSCs from the tumor microenvironment can also transdifferentiate into M2-type macrophages, myeloid-derived suppressor cells (MDSC) or M2-type macrophages under the influence of cytokines or chemokines [75,76,77].

## 3. Stroma Phenotype Associated with Tumor Metastasis

During the past 10 years, our group has investigated the relationship of different factors expressed by stromal cells associated with breast cancer metastasis. These factors include matrix metalloproteases (MMPs) and their tissue inhibitors (TIMPs), cytokines and Toll-like receptors (TLRs). Based on these data, we can propose two types of breast carcinoma stroma with significantly different influences on prognosis (Figure 1).

### 3.1. Metalloproteases and Their Inhibitors

The human MMP family is composed by 26 members divided into six groups based on substrate specificity and homology (Table 1). MMPs play an essential role in the degradation of the stromal connective tissue and basement membrane components, which are key elements in tumor invasion and metastasis. MMPs are also able to influence in vivo tumor cell behavior due to their capacity to cleave growth factors, cell surface receptors, cell adhesion molecules and chemokines/cytokines [78,79]. Furthermore, by cleaving proapoptotic factors, MMPs produce a more aggressive phenotype through the generation of cells resistant to apoptosis [78]. MMPs also positively regulate cancer-related angiogenesis, through their ability to mobilize or activate proangiogenic factors (bFGF, VEGF, TGFβ and integrin αvβ3) after the degradation of basement membrane or ECM components and negatively via generation of angiogenesis inhibitors, such as angiostatin and endostatin [80]. MMP activity is specifically inhibited by TIMPs, but it is now accepted that TIMPs are multifactorial proteins also involved in the induction of tumor cell proliferation and the inhibition of apoptosis [81,82]. The role of MMPs in ECM remodeling is mainly due to their capacity to degrade the ECM, thus allowing tumor progression. In this sense, MMP-7 (matrilysin 1), with the capacity to degrade type IV collagen, fibronectin and laminin, is highly expressed in human breast tumors, inducing tumor growth and invasiveness [83]. MMP-9 (gelatinase B) is related to tumor invasion and metastasis by its special capacity to degrade type IV collagen [84] and to induce angiogenesis [78]. MMP-11 (stromelysin-3) is preferentially expressed by peritumoral stromal cells and is associated with tumor progression and poor prognosis [85,86,87]. MMP-13 (collagenase-3), due to its wide substrate specificity compared with other MMPs, plays a central role in the MMP activation cascade [88]. MMP-14 (membrane type 1 MMP or MT1-MMP) is involved in ECM degradation, activation of MMP-13 and MMP-2 zymogen, and in molecular carcinogenesis, tumor cell growth, invasion and angiogenesis.

Breast carcinomas containing mononuclear inflammatory cells (MICs) or CAFs with a high expression profile of MMPs and TIMPs have a higher rate of distant metastasis development compared with tumors with a low expression profile [86,87,89,90], independently of luminal or basal-like phenotype of breast carcinomas [91]. However, variations in MMP/TIMP expression among the different histological subtypes of breast carcinomas (ductal, lobular, mucinous, tubular, papillary and medullary invasive carcinomas) have been found [92]. In a more recent study, we found that MMP-11 (also known as stromelysin 3) expression by MICs, and TIMP-2 expression by CAFs, either at the tumor center or at the invasive front, were the most potent independent prognostic factors for predicting the clinical outcome of patients [87]. Considering that the expression of MMP-11 may constitute a useful biological marker for pro-metastatic MICs, we investigated its relationship with the gene expression of 65 factors associated with inflammation and tumor progression in a population of breast cancer patients stratified into two groups according to MMP-11 expression by intratumoral MICs (positive or negative). Among all analyzed factors, interleukin 1β (IL-1β), IL-6, IL-17, interferon β (IFNβ) and nuclear factor kappa B (NFκB) were expressed at high levels in tumors with MMP-11 positive MICs [93,94]. It has been evidenced that MMPs can either promote or repress inflammation by the direct proteolytic processing of cytokines to activate, inactivate, or antagonize their function. For example, IL-1β requires proteolytic processing for activation. Indeed, the IL-1β-converting enzyme (ICE, nowadays known as caspase-1) needs the activity of MMPs, such as MMP2, -3, and -9, to activate the IL-1β precursor to the active form [95].

We also found that MMP/TIMP expression by endothelial cells (ECs) from the adjacent non-neoplastic tissue was absent or very low compared with ECs from the tumor itself. In addition, MMP-11 expression by ECs was related to distant metastasis development and shorter relapse-free survival, whereas, conversely, TIMP-3 expression was related to low occurrence of distant metastasis [96]. These results support our previous data indicating that the expression of MMP-11 by stromal cells is associated with distant metastasis development in breast cancer. Regarding TIMP-3, there are data in agreement with our own findings indicating that TIMP-3 is a naturally occurring inhibitor of angiogenesis that limits vessel density in the vascular bed of tumors and curtails tumor growth [97,98]. In addition, it has been reported that TIMP-3 may induce apoptosis in ECs by triggering a caspase-independent cell death pathway [99].

### 3.2. Cytokines

Cytokines are low-molecular-weight proteins that mediate cell-to-cell communication. Immune and stromal cells, such as fibroblasts and endothelial cells, synthesize cytokines and regulate through them several processes, such as proliferation, cell survival, differentiation, immune cell activation, cell migration and death. Besides the central role of cytokines in the inflammatory process, they have also been recognized as powerful players in tumor progression through several pathways, including the generation of free radicals that can damage DNA, potentially causing mutations that lead to tumor initiation, stimulating cell proliferation and reducing apoptosis, stimulating EMT and angiogenesis or allowing tumor cell evasion of immune surveillance. On the other hand, cytokines can modulate an anti-tumoral response that seems conditional on the balance of pro- and anti-inflammatory cytokines, their relative concentrations, cytokine receptor expression, the activation state of surrounding cells [100] and the stage of tumor development [101].

Although previous reports have shown several positive associations between high cytokine levels and tumor aggressiveness, most of them were based on serum levels and few of them evaluated the impact on tumor recurrence [102,103,104]. We have reported high cytokine expression by cancer cells, supporting the recognized fact that cancer cells secrete cytokines that can act as autocrine factors contributing to their malignant phenotype. We also found different profiles of cytokine expression by stromal cells related with patient outcome [105]. Indeed, IL-1β expression by stromal cells (CAFs and MICs) was significantly associated with both low metastasis rate and longer relapse-free survival and overall survival, whereas IL-10 expression by stromal cells was significantly associated with higher metastasis occurrence and both shorter relapse-free survival and overall survival. In addition, the combination of IL-1β, IL-6 and IL-10 expression by MICs showed an important association with prognosis and improved the prognostic significance of MMP-11 status by MICs [105]. All these data seem to indicate that the non-expression of pro-inflammatory cytokines (such as IL-1β and IL-6), together with the expression of an anti-inflammatory cytokine (such as IL-10) could contribute to tumor immune escape. IL-10 is an important anti-inflammatory cytokine and due to its immunosuppressive effect on dendritic cells and macrophages, IL-10 can dampen antigen presentation, cell maturation and differentiation, allowing tumor cells to evade immune surveillance mechanisms [106]. These findings are in agreement with some studies suggesting that interleukins can inhibit tumor growth [107,108] and that their expression can be correlated with good prognosis [109,110]. All of these data suggest the complexity of the tumor stroma and the importance to consider the particular cell type expressing cytokines in the context of the tumor environment.

### 3.3. Toll-Like Receptors (TLRs)

TLRs play an essential role in the activation of innate and adaptive immunity, contributing to the capacity of the immune system to combat pathogens [111]. Cancer cells activated by TLR signals may also release cytokines and chemokines that in turn may recruit immune cells and stimulate them to release further cytokines and chemokines, resulting in a cytokine profile associated with immune tolerance, tumor cell proliferation and resistance to apoptosis; but it also enhances tumor cell invasion and metastasis by regulating metalloproteases and integrins [112,113,114]. However, there are data supporting the importance of TLR expression by stromal cells in tumor behavior. In this sense, we have shown that TLR4 expression by MICs was associated with an increased incidence of metastasis, whereas TLR9 expression by CAFs was associated with a low metastasis-rate [115], pointing towards a protective role of TLR9 against tumor progression. Indeed, it has been shown that stimulation of TLR9 activates human plasmacytoid dendritic cells and B cells, inducing a potent innate immune response, in preclinical tumor models as well as in patients [116]. TLR9 ligands induce the expression of OX40 on CD4^+^ T cells in the tumor microenvironment. OX40, a co-stimulatory molecule belonging to the TNFR superfamily expressed on activated effector T cells and regulatory T cells, can promote effector T cell activation and inhibit regulatory T cell function, which may contribute to the eradication of spontaneous malignancy by local immunotherapy [117].

### 3.4 Integrins

Cell adhesion to the ECM is fundamental for tissue integrity and basic behavioral responses of cells under physiological conditions. Integrins are the main cell adhesion receptors for components of the ECM. The name integrin refers to the function of family members to integrate cell exteriors (e.g., ECM) to the cell interiors (e.g., the cytoskeleton) [118]. They are a family of 24 transmembrane heterodimers generated from a combination of 18α integrin (with a size of 120–170 kDa) and 8β integrin (with a size of 90–100 kDa) chains, connected by monovalent bonds. Both α and β subunits possess a large extracellular domain, a transmembrane domain and usually a small cytoplasmic tail [119]. Thus, both integrin subunits are required for interactions with the cytoskeleton and the ECM [120]. Integrins markedly differ from each other in ligand specificity and expression levels in mammalian tissues. The most common and vital for mammalian cells are fibronectin-binding integrin α5β1, the collagen-binding receptor α2β1, and integrin αvβ3, having a diverse ligand specificity.

The role of integrins, primary receptors for ECM and bi-directional signaling molecules, is complex because they bind to some matrix proteins and to other receptors at the same time. Integrins act as sensors of the epithelial microenvironment by affecting cells in two key ways: regulating the actin cytoskeleton of cells through binding directly with proteins (filamin, talin and vinculin) [121], or by phosphorylating the relative kinases (focal adhesion kinases (FAKs), proto-oncogene tyrosine-protein kinase (Src)-family kinases (SFKs) and integrin-linked kinase (ILK), to activate or cooperate with the other cell signaling pathways including the PI3K/Akt and MAPK/Erk pathways [122,123]. Thus, they transduce information (“outside-in”) from the extracellular environment to modulate cell responses, including survival signaling, growth signaling, adhesion, spreading, migration, secretion of proteases and invasion [124,125]. For all of these reasons, integrins do not just bind a cell to its environment, but they have also been shown to regulate several intracellular signaling pathways, rendering their physiological role far more complex [118]. Indeed, dysregulated integrin-mediated adhesion and signaling is a precursor in the pathogenesis of many human diseases, including cancer [126].

## 4. Intercellular Communication System in the Tumor Microenvironment

The bi-directional signaling between cancer cells and stroma, mainly composed by CAFs and MICs, induces the expression of pro-tumoral factors by both tumor compartments [127,128,129]. These signals can be mediated by soluble factors, exosomes or via integrins.

### 4.1. Soluble Factors

Cancer cells secrete cytokines and chemokines, such as TGF-β, involved in the recruitment and activation of CAFs [128,130]. Furthermore, cytokines and chemokines secreted by cancer cells and CAFs, such as CCL2, contribute to the recruitment of macrophages and the induction of their transformation into tumor-associated macrophages (TAMs) [131,132,133]. In this sense, it has been reported that the oncogenic dysregulation of the RAS, MYC and the MAPK pathways in cancer cells contributes to the secretion of growth factors and cytokines such as VEGF, IL-6, IL-10, and IL-1β, leading to the recruitment and the tumorigenic transformation of macrophages [134,135].

Nevertheless, after the recruitment of stromal cells in the tumor microenvironment, a complex dynamic interaction takes place, which can be evidenced by transwell-type experiments. Thus, it was demonstrated that after co-culture of CAFs and breast cancer cell lines (both MCF-7 and MDA-MB-231 cell lines), gene expression of factors related to tumor progression was significantly upregulated in both cell types (Figure 2) and invasion and angiogenic capacities of breast cancer cells were increased. Among all analyzed factors, S100A4, FGF7, PDGFB, VEGFA, TGFβ, IL6, IL8, uPA, MMP2, MMP11 and TIMP1 are the more differentially expressed (Table 2) [13,136]. Additionally, the expression of some of these factors [13,136] was highly increased after co-culture of breast cancer cell lines with CAFs from the primary tumor and MIC-MMP11+ cells [136], which suggests a strong pro-tumor influence of the immune original microenvironment of the CAFs.

### 4.2. Exosomes

Although soluble factors, chemokines and cytokines produced within the tumor are mainly responsible of modifications in cancer cells and stromal cells, new evidence indicates the contribution of another mode of cell communication involving extracellular vesicles (EVs). EVs, resulting in a particulate nano-communication system that may be responsible for dissemination of messages among the different cell types of the tumor, are divided into different categories depending on their size: apoptotic bodies (1000–5000 nm), microvesicles (500–1000 nm in diameter) and exosomes (30–150 nm) [137]. Exosomes, the smallest subset of EVs, originate in the endocytic compartment of the parent cell via a series of intraluminal invaginations taking place in the multivesicular bodies (MVBs). Consequently, their molecular content recapitulates the content of the parent cell, at least partially [138]. Exosomes are enclosed by a protein-phospholipid bilayer membrane interspersed by cell type-specific proteins, lipids and glycans. The exosome lumen is filled with various cellular proteins, nucleic acids, mRNA, miRNA and DNA, soluble factors, including cytokines and chemokines, enzymes and cofactors [139]. Exosomes produced by different cell types carry distinct molecular and genetic components, and they may be “addressed” by the parent cell to reach a specific molecular address of the recipient cell. Upon contacting a local or distantly-located recipient cell, exosomes deliver signals that culminate in cellular re-programming [140,141]. The mechanisms responsible for delivery and processing of the exosome cargo in recipient cells are not entirely understood, but may include the initial ligand-receptor type of binding on the cell surface followed by endocytosis or phagocytosis of exosomes [142].

### 4.3. Tumor-Derived Exosomes (T-D-EXs)

Tumor cells are important producers of exosomes with molecular signatures depending on the type of tumor cell, which are different from those of exosomes derived from normal cells [143,144]. Tumor-derived exosomes (T-D-EXs), ubiquitously present in the tumor environment and in body fluids of patients with cancer [58,143], circulate and disseminate information to tissues and cells distant from the tumor. T-D-EXs carry messages from the parent tumor cell to other normal or malignant cells in the tumor microenvironment, including MSCs [145]. Thus, T-D-EXs can mediate autocrine, juxtacrine and paracrine signaling that the tumor cells establish and that is necessary for their survival in the tumor microenvironment [146] (Figure 3).

### 4.4. MSC-Derived Exosomes (MSC-D-EXs)

Under normal physiological conditions, MSCs are a rich source of exosomes [64], which seem responsible for many functions generally attributed to MSCs [147], such as their contribution to the modulation of physiological functions of neighboring stromal cells [148]. It is known that MSC-derived exosomes (MSC-D-EXs), through spontaneous or organelle-mediated release, significantly contribute to the increase in extracellular ATP levels [149], which induces a favorable effect in migration and invasion of breast cancer cells [150].

Importantly, MSC-D-EXs have the capacity to interact with multiple cell types in the tumor microenvironment and to ensure they adequately support tumor growth. MSC-D-EXs carry a complex cargo of molecules and genes comprising more than 850 unique gene products and more than 150 different miRNAs [151,152] and thus have the potential to elicit different cellular responses in a broad variety of cells [153]. Nevertheless, MSCs are recognized as recipients of signals emanating from the tumor, but also as efficient producers of their own and abundant exosomes [66,153]. Thus, exosomes horizontally transfer information to neighboring cells and transform the cellular milieu to one supporting tumor survival [153]. (Figure 3). It is known that T-D-EXs induce phenotypic and functional changes in MSCs which may exert profound effects on tumor growth [154], and that epigenetic modifications mediated by genetic elements introduced by exosomes to recipient MSCs appear to be involved [155]. At present, the cellular, molecular and genetic mechanisms responsible for re-programming of MSCs by T-D-EXs are under intense scrutiny. The initial contact may be due to T-D-EXs carrying numerous cell adhesion molecules (CAMs) whereby they can readily fuse with adhesion receptors on MSCs allowing for the protein/gene transfer to the MSCs cytosol. It remains unclear whether the protein transfer alone is sufficient for the re-programming of MSCs by T-D-EXs or whether the transfer of transcription factors and nucleic acids is mandatory. These changes in MSCs include the overexpression of genes involved in cell migration (CXCR4 and CXCR7), in the matrix remodeling (collagen type IV alpha 3 chain) and in angiogenesis or tumor growth (IL-8, OPN and myeloperoxidase) [156]. T-D-EXs from prostate cancer, breast cancer or chronic lymphocytic leukaemia can promote MSC migration to the tumor site [55] and induce MSC differentiation into myelofibroblasts overexpressing alpha smooth muscle actin (αSMA) [65].

### 4.5. Integrins and Cancer

Alterations of integrin expression and function, linked to many types of cancer [157,158], lead to modifications which are the basis of tumor progression, such as dedifferentiation, proliferation, apoptosis or disorganization of the ECM and promotion of metastasis [122,157,159,160,161,162,163]. Therefore, integrins have been implicated in different steps of cancer progression, such as cancer initiation and proliferation, local invasion and intravasation, survival of circulating tumor cells (CTCs), extravasation and metastatic colonization [163].

Regarding breast cancer, experimental data implicate integrins such as β1 integrins, and more specifically α3β1 integrin, in mammary tumorigenesis [164,165]. Likewise, aberrant expression of β1 integrin in human breast carcinoma has been linked to cell adhesion, angiogenesis, tumor progression and metastasis [166,167,168]. In agreement with these findings, there are clinical data demonstrating that integrin overexpression, such as that of β1 integrin [167,168,169] or integrin α6 [170], are associated with a poor prognosis and reduced survival, or with radiation resistance [171]. Nevertheless, the functions of individual integrins in invasion and metastasis processes are controversial. Thus, for example, it has been reported that inhibition of β1 integrin significantly reduces the formation of metastatic foci of several cancer types, including breast cancer [172]; or that α2β1 suppressed metastatic dissemination in a mouse model of spontaneous breast cancer [173]. In addition, some studies have reported that decreased β1 integrin protein expression is associated with more aggressive breast cancer types [174,175], whereas other studies could not verify a significant correlation between integrin subunit β1 (ITGB1) protein expression and survival of patients with breast carcinoma [176,177]. These contradictory data may be due to the diversity of the integrin family and the variety of signal pathways that a particular receptor can induce in different cells, as well as to differences between the different tumor histotypes in signaling pathways initiated by integrins. Therefore, it may be of key importance to consider the role of integrins in the context of tumor stroma heterogeneity, representing an exciting new way to explore the complexity of biological systems.

### 4.6. Possible Role of Integrins to Better Characterize the Tumor Stroma Phenotype in Breast Cancer

Integrins are widely expressed by many types of cells, including tumor cells, endothelial cells (ECs), pericytes, fibroblasts and immune cells. Therefore, integrins play key roles in other cancer-relevant processes, such as white blood cell trafficking and activation, chronic inflammation and angiogenesis, which are strongly related to cancer progression [163]. The contribution of CAFs to cancer progression and tumor invasion is mediated via several integrin-linked mechanisms [178,179]. CAFs align fibronectin fibres within the tumor ECM and, through application of traction forces mediated by α5β1 integrin, promote directional cancer cell migration [180]. Additionally, CAFs are able to induce tamoxifen resistance by secreting fibronectin, which stimulates its ligand integrin β1 to activate signaling pathways, such as the PI3K/AKT pathway [181] and long-term exposure to CAFs makes breast cancer cells addictive to integrin β1 [182].

Although integrin-mediated cell-to-cell and cell-to-matrix interactions during the T-cell lifespan still represent an open field of research, there are data indicating that integrins in lymphocytes or macrophages also play a key role to promote tumor progression. Thus, several mechanisms implicate integrins in the infiltration of lymphocytes and macrophages into tumors. This may be by a dynamic integration of integrins with several extracellular factors, such as the ECM protein periostin [183], osteopontin-rich matrix [184], VCAM-1 [185,186] or fibronectin [185]. In addition, it has been reported that αLβ2 affinity down-modulation is crucial in promoting the intravascular crawling and diapedesis of T-cells during homing to peripheral lymph nodes [187]. Integrins also generate a signal that interacts with chemokines and antigens to modulate T-cell motility, proliferation and differentiation [188]. Endothelial cells use integrins to interact with their underlying basement membrane, which combined with tyrosine kinase-induced signaling are important regulators of vessel integrity and tumor progression, since integrin expression both on cancer cells and on endothelial cells is implicated in extravasation [189].

Interestingly, integrins also regulate proteins, such as MMPs and fibronectin, which play key roles in the dynamics of the proteolytic activity from tumor stroma and the metastasic cascade. Integrins contribute to the latter by upregulating the expression of MMP genes and facilitating protease activation and function at the ECM interface [190]. In addition, integrins, such as αVβ3 cooperate with MMPs, such as MMP-2 [191], MMP-9 [192] and MMP-14 [193], in regulating migration of metastatic breast cancer cells toward specific substrates in an activation-dependent pathway. Indeed, it has been demonstrated that integrins control MMP-2 and MMP-9 expression regulating angiogenesis in breast tumor cells and endothelial cells [194]. In fact, the inhibition of MMP-9 and α_V_β_5_-integrin interaction results in a reduced angiogenesis and tumor invasion [195].

The arginine (R)-glycine (G)-aspartic acid (D) (RGD) sequence is included in the adhesion molecule fibronectin, which is the ligand for several types of integrins, such as α5β1, αvβ3 and αvβ5 [124]. Integrins, after binding to fibronectin, activate focal adhesion kinase (FAK), which further activates multiple signaling proteins, promoting directional cancer cell migration through the activation of cytoskeletal contractility [178,180]. It is also worth mentioning that integrin-fibronectin interactions are also implicated in induced exit of cancer cells from dormancy [196,197], as well as in radiotherapy resistance [198,199]. As was mentioned above, there are new data which implicate exosomes in playing a vital role in the development of organ-specific metastasis [200], integrins being the most highly expressed receptors on their surface. These associations might contribute to support the “seed and soil” hypothesis proposed by Paget more than 100 years ago [201]. This is because integrins facilitate the binding and fusion of extracellular vesicles to the plasma membrane of their cell targets [202]. Thus, exosomal integrins can contribute to initiate organ colonization of specific tissues by preparing favorable pre-metastatic niches to metastatic niche formation [200,203,204,205]. With regard to this, it has been shown that integrins αvβ5, α6β4 and α6β1 on tumor-derived exosomes could drive tumor cells to metastasize to specific organs like lung, liver or brain [203].

### 4.7. Tumor Stroma and Therapeutic Opportunities

A consolidated example of targeting stroma in breast cancer are the different strategies available to target angiogenic cells in clinical trials of advanced breast cancer [206]. In addition, cancer therapies should aim for a progressive disruption of the dynamics of interactions between cancer cells and the tumor microenvironment by targeting metabolic dysregulation and inflammation to partially restore tissue homeostasis and turn on the immune cancer kill switch. The translation of this therapeutic approach to established treatments would, however, require more understanding of the adaptive complexity of cancer resulting from the interactions of cancer cells with the tumor microenvironment and the immune system.

Studies investigating the role of CAFs have reported that the therapeutic targeting of cancer cells alone is not enough for the treatment of cancer [207]. CAFs are essential components of the tumor microenvironment and therefore, represent a molecular target for the treatment of cancer [208]. In addition, compared with cancer cells, CAFs are genetically more stable with a reduced probability of developing drug-resistance, thus representing a potential therapeutic target with a lower probability of long-term chemoresistance development [209,210]. Different strategies implicating CAFs could be developed, such as targeting the ability of CAFs to exert mechanical forces on the basal membrane [6], or induce the reduction of lactate and steer the tumor microenvironment to a state of reduced inflammation so as to enable an effective intervention of the immune system. This is because the probability of dysregulation of the RTK, PI3K and MAPK signaling pathways is significantly high for most types of cancer. Driven by growth factors from the stroma, these pathways may, with high probability, be the first drivers of an upregulated glycolysis in cancer cells. The consequent increase of lactate secretion into the tumor microenvironment will thereafter lead to its acidification and the activation of TGF-β [211], leading to the recruitment and transformation of CAFs. On the other hand, several new agents blocking CAF pro-tumor activity have undergone pre-clinical and clinical evaluation [12,212,213]. Several clinical trials have also been implemented in order to evaluate inhibitors of cytokine receptors or neutralizing antibodies that prevent the sustained exposure to these inflammatory mediators that promote tumor progression [214,215]. Among the most implemented clinical trials for breast cancer in the metastatic setting are those involving drugs that target immune-cell-intrinsic checkpoints. Blockade of one of these checkpoints, cytotoxic T-lymphocyte-associated antigen-4 (CTLA-4) or the programmed cell death 1 (PD-1) receptor, may provide proof of concept for the activity of an immune-modulation approach in the treatment of breast cancer [216]. As it was previously reported, MSC biology is of great interest for cancer progression and could lead to new therapeutic strategies. Thus, for example, it has recently been reported that the trophic effect of MSCs on breast cancer cell growth is exerted via ionotropic purinergic signaling, suggesting the inhibition of the purinergic signaling system as a potential target for therapeutic intervention [217]. Nevertheless, contradictory results have been described regarding induced pro- or anti-tumor effects, both in vitro and in vivo [218]. These discrepancies in MSC functionality may also be due to their heterogeneity and to the fact that no specific surface markers currently exist to isolate a more homogeneous population. Some data point towards the importance of the source of MSCs with regard to their tumorigenic properties. Thus, for example, bone marrow-derived MSCs (BM-MSC-CM) have both anti-tumor effects on non-small cell lung cancer cells [219] and stimulatory effects on myeloma cells [220], whereas adipose tissue-derived MSCs conditioned medium (ADSC-CM) has no effect on human glioblastoma cancer stem cell subpopulations [221]. Therefore, a possible alternative would be to find an MSC-tissue source providing a unique anti-tumor activity. Regarding this aspect, we recently identified the human cervix as a new source of MSCs, named human uterine cervical stem cells (hUCESCs), which are obtained from the transformation zone of the uterine cervix of healthy women by means of Pap cervical smears [222]. Our results show specific anti-tumor effects of conditioned medium from hUCESCs (hUCESCs-CM) on proliferation, apoptosis, tumor-cell invasiveness and aggressive behavior of breast cancer cells in vitro (MDA-MB-231 breast cancer cell lines and primary tumors) and reduced tumor growth in vivo in a mouse xenograft tumor model, which differ from those reported for other types of MSC-CM [223,224,225]. We also explored the effect of hUCESCs-CM treatment on CAFs and on monocyte to macrophage differentiation [222]. In this sense, we established that co-culture with hUCESCs as well as treatment with hUCESCs-CM significantly reduced CAF cell proliferation and invasion, and increased early apoptosis. We also observed that treatment with hUCESCs-CM significantly inhibited and reversed macrophage differentiation. These are interesting findings, because the use of secretome and conditioned medium from MSCs is considered a new therapeutic strategy [226].

The secretome, emerging as a novel type of biological regulation involving the communication between cells, is defined as the set of factors/molecules secreted to the extracellular space. These factors include, among others, soluble proteins, free nucleic acids, lipids and extracellular vesicles including exosomes. The use of cell-free therapies such as MSC-sourced secretome in regenerative medicine provides key advantages over stem-cell based applications [226]: its application resolves several safety considerations potentially associated with the transplantation of living and proliferating cell populations (immune compatibility, tumorigenicity, emboli formation and the transmission of infections); it may be evaluated for safety, dosage and potency in a manner analogous to conventional pharmaceutical agents; its storage can be performed without application of potentially toxic cryopreservative agents for a long period without loss of product potency; it is more economical and more practical for clinical application since it avoids invasive cell collection procedures; its mass-production is possible through tailor-made cell lines under controlled laboratory conditions; etc. Our study revealed that hUCESCs-CM had a higher concentration than control media or ASCs-CM of factors such as LIGHT (or TNFSF14), Fms-related tyrosine kinase 3 ligand (FLT-3 ligand), interferon-gamma-inducible protein-10 (IP-10) and latency-associated protein [222]. These factors have been associated with induction of apoptosis, inhibition of cell growth, reduced cell invasion and tumor inhibition. It is also noteworthy that, compared to ASCs-CM, hUCESCs-CM contains lower levels of factors known to participate in cancer progression, such as epidermal growth factor receptor (EGFR), fibroblast growth factor (FGF) 4 and 9, Intercellular adhesion molecule 3 (ICAM3), IL6, IL6R, monocyte chemotactic protein-3 MCP3 (also named CCL7), macrophage migration inhibitory factor (MIF), sgp130 and vascular endothelial growth factor D (VEGFD) [222]. This, again, indicates that the properties of MSCs and their secretomes vary depending on their source. All of these findings related to the anatomical source of hUCESCs [227], suggest that a complex paracrine signaling network is implicated in the anti-tumor potential of this MSC type. In addition, recent in vitro data of our group demonstrate that exosomes derived from hUCESCs show an anti-tumor effect (unpublished data), which is interesting because of the potential for future therapeutic interventions for cancer control via MSCs-derived exosomes.

On the other hand, integrins may be interesting therapeutic targets because they are expressed by stromal cells from primary tumors and from metastatic niches (CAFs, immune-cells or endothelial cells) and on tumor-derived exosomes [163]. In fact, previous studies have demonstrated that blocking integrins with synthetic peptides, antibodies, or disintegrins, interferes with tumor cell invasion and metastasis in vitro and in vivo [228,229]. Nevertheless, there is still need for reliable predictive biomarkers for patient stratification, as well as the correspondent detection tools, such as antibodies which reliably function in immunohistochemistry performed on formalin-fixed, paraffin-embedded material [229].

Their development would be a major step forward in the integrin field to translate preclinical efficacy into a beneficial treatment for patients.

## 5. Conclusions and Future Perspectives

Studies on the cellular components of breast cancer stroma have identified new biological markers, such as MMPs, TIMPs, cytokines or TLRs leading to the definition of main phenotypes with different prognoses. These and other factors, such as exosomes derived from cancer cells, MSCs and integrins, could allow us in the near future to identify pathways of intercellular communication leading to new therapeutic strategies in breast cancer.

Studies investigating the role of tumor stromal cells have reported that the therapeutic targeting of cancer cells alone is not enough for the treatment of cancer. There is evidence indicating that the tumor microenvironment is a fertile ground for the development of novel therapies with the potential to augment existing treatment and prevention options. In addition, compared to cancer cells, stromal cells are relatively stable from the genetic point of view, with a reduced probability of developing drug-resistance, thus representing a potential therapeutic target with lower chances for the development of long-term chemoresistance. Several clinical trials have been implemented in order to evaluate blocking CAFs or inhibitors of cytokine receptors or neutralizing antibodies that prevent the sustained exposure to inflammatory mediators that promote tumor progression. MSC biology is also of great interest in cancer progression and could lead to new therapeutic strategies. A possible alternative would be to find an MSC-tissue source providing a unique anti-tumor activity. With regard to it, hUCESCs have shown a wide anti-tumor effect against aggressive breast cancer in both in vitro and in vivo studies, against CAFs and on monocyte to macrophage differentiation in breast cancer. CM or secretome from hUCESCs emerges as an interesting and potential anticancer cell-free therapy in regenerative medicine, providing key advantages over stem-cell based applications, resolving several safety considerations potentially associated with the transplantation of living and proliferative cell populations, and carrying several technical advantages for its application. Finally, considering that integrins are key players in metastasis and are expressed by stromal cells from primary tumors and from metastatic niches and on tumor-derived exosomes, targeting integrins may be an important innovative therapy to explore when considering future therapeutic options based on tumor stroma activity blockage.

## Figures and Tables

**Figure 1 cancers-11-00664-f001:**
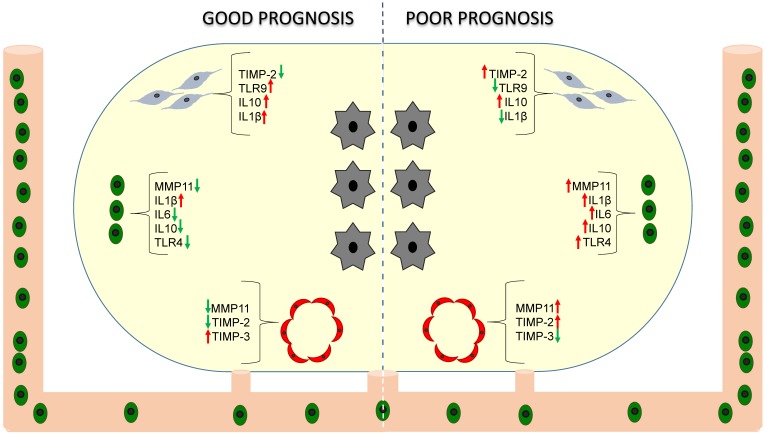
Stroma phenotype and prognosis. Prognostic significance of factors expressed by cancer-associated fibroblasts (CAFs, 
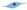
), mononuclear inflammatory cells (MICs, 
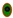
) and endothelial cells (
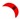
) in breast cancer (tumor cells, 
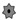
).

**Figure 2 cancers-11-00664-f002:**
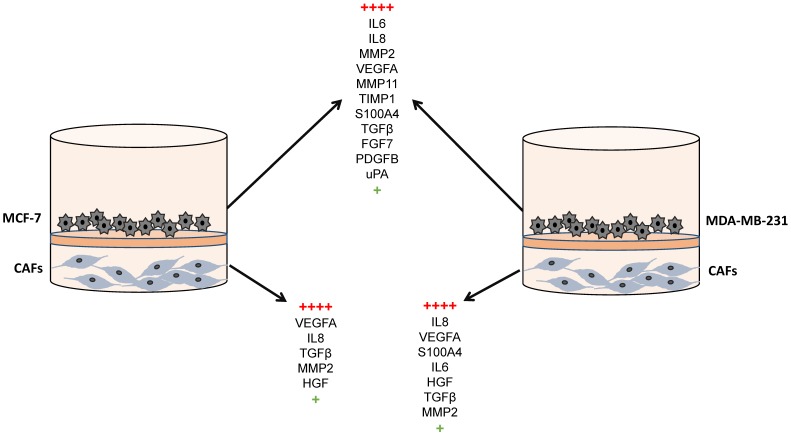
Gene expression of factors after co-culture between CAFs and breast cancer cell lines (MCF-7 or MDA-MB-231).

**Figure 3 cancers-11-00664-f003:**
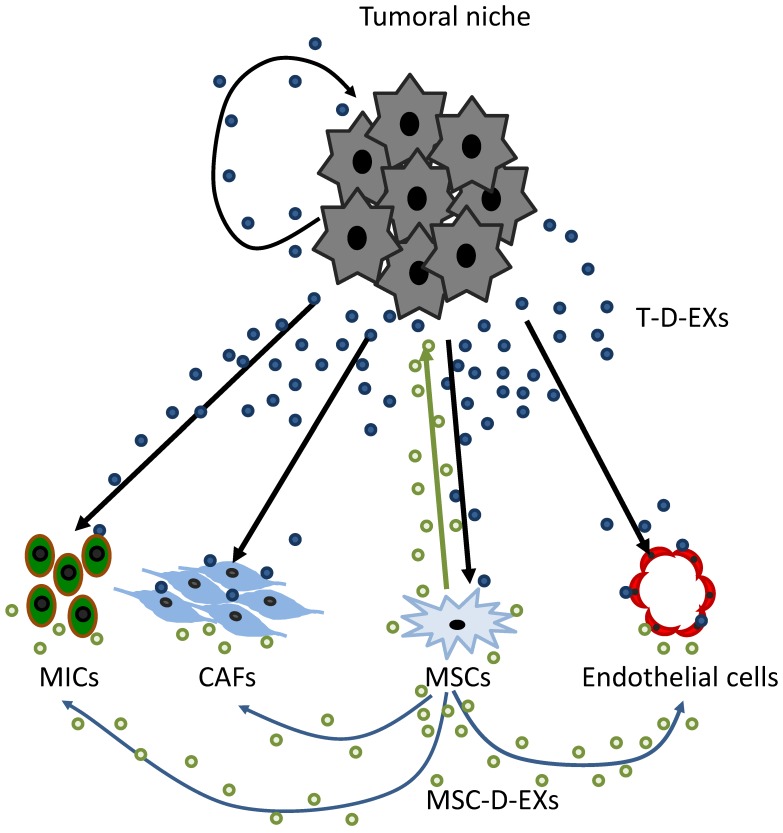
Paracrine communication between cancer cells and tumor microenvironment through exosomes.

**Table 1 cancers-11-00664-t001:** Human matrix metalloproteases.

Name of Class	MMP	Enzyme Name	Substrates
Collagenases	MMP-1	Collagenase-1	Collagens (I, II, III, VII and X), proteoglycans, entactin, ovostatin, MMP-2, MMP-9
	MMP-8	Collagenase-2/neutrophil collagenase	Collagens (I, II, III, VII, VIII and X), fibronectin, proteoglycans
	MMP-13	Collagenase-3	Collagens (I, II, III, VII, VIII and X), tenascin, plasminogen, aggrecan, fibronectin, osteonectin, MMP-9
	MMP-18	Collagenase-4	Type I collagen
Gelatinases	MMP-2	Gelatinase-A	Gelatin, collagen (IV, V, VII VI, IX and X), elastin, fibronectin
	MMP-9	Gelatinase-A	Collagens (IV, V, VII, X and XIV), gelatin, entactin, elastin, fibronectin, osteonectin, plasminogen, proteoglycans
Stromelysins	MMP-3	Stromelysin-1	Collagens (IV, V and IX), gelatin, aggrecan, laminin, elastin, casein, osteonectin, fibronectin, ovostatin, entactin, plasminogen
	MMP-10	Stromelysin2	Collagens (I, II, IV and V), gelatin, casein, elastin, fibronectin
	MMP-11	Stromelysin2	Collagens (IV, V, IX and X), laminin, elastin, fibronectin, casein, proteoglycans
	MMP-17	Homology tostromelysin-2	Pro-MMP2, fibrin/fibrinogen, gelatin
Matrilysins	MMP-7	Matrilysin	Collagens IV, gelatin, fibronectin, laminin, elastin, casein, transferrin
	MMP-26	Matrilysin-2	Collagen IV, fibronectin, fibrinogen, gelatin, pro-MMP9
MT-MMP (membrane type-MMP)	MMP-14	MT1-MMP	Collagens (I, II, III), gelatin, fibronectin, laminin, vitronectin, entactin, pro-MMP2
	MMP-15	MT2-MMP	Fibronectin, gelatine, vitronectin, entactin, laminin, pro-MMP-2
	MMP-16	MT3-MMP	Collagen III, gelatin, casein, fibronectin, pro- MMP-2
	MMP-17	MT4-MMP	Pro-MMP2, fibrinogen, gelatin
	MMP-24	MT5-MMP	Fibronectin, pro-MMP2, proteoglycans, gelatin
	MMP-25	MT6-MMP	Pro-MMP2, pro-MMP9, collagen IV, gelatine, fibronectin, Proteinase A
Other enzymes	MMP-12	Macrophage metalloelastase	Collagen IV, gelatin, elastin, casein, fibronectin, vitronectin, laminin, entactin, fibrin/fibrinogen
	MMP-19	RASI-1	Collagen (I, IV) gelatin, fibronectin, laminin
	MMP-20	Enamelysin	Amelogenin, aggrecan
	MMP-21		
	MMP-22		
	MMP-23		Gelatin
	MMP-28	Epilysin	
	MMP-29	Unnamed	

**Table 2 cancers-11-00664-t002:** Main role of factors implicated in the crosstalk between cancer cells and CAFs.

Gene Symbol	Gene Name	Main Role
S100A4	S100 calcium binding protein A4	Invasion
FGF7	Fibroblast growth factor 7	Cell growth/Invasion
PDGFB	Platelet-derived growth factor beta	Angiogenesis
VEGFA	Vascular endothelial growth factor A	Angiogenesis
TGFβ	Transforming growth factor beta	Inflammation/Invasion
IL6	Interleukin 6	Inflammation
IL8	Interleukin 8	Inflammation
uPA	Urokinase-type plasminogen activator	ECM remodeling
MMP2	Matrix metalloproteases 2	ECM remodeling
MMP11	Matrix metalloproteases 11	ECM remodeling
TIMP1	Tissue inhibitor of metalloproteases 1	ECM remodeling

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
