# Peer review of "Breast Cancer Tumor Stroma: Cellular Components, Phenotypic Heterogeneity, Intercellular Communication, Prognostic Implications and Therapeutic Opportunities"

_cancers, 2019, doi:10.3390/cancers11050664_

Round 1
Reviewer 1 Report
This is a very nice review of an important topic. There are minor spelling corrections that are needed such as "Cellular" and "Communication" in the title; but for the most part the English is excellent. The review seems quite comprehensive and unbiased. I think it might help clinicians and other non-basic scientists to understand some potential new targets that could be brought to clinical trial. I would recommend publication after minor spelling errors are corrected.
Author Response
Reviewer 1
There are minor spelling corrections that are needed such as "Cellular" and "Communication" in the title
As suggested by the Referee, we have corrected spelling mistakes in the title.
Reviewer 2 Report
This review addresses the role of the stroma in breast cancer. The text is mostly well written and explores pertinent aspects of the stroma in carcinogenesis.
1. Is there information on stroma specifics in breast cancer subgroups, ie BRCA, ER+, PR+ HER2+, ductal, lobular etc?
2. MSC is mostly referred to as mesenchymal stem cells but actually in the vast majority of studies these cells have not been demonstrated as true stem cells. “Mesenchymal stromal cells” is more appropriate usage unless the cells in question have been demonstrated to be true stem cells or at least this distinction should be clarified. This applies to the text throughout the manuscript.
Minor points
Lines 24-25 and 48-49: “autonomous progression” and “cellular sociology” I find dubious from both the perspectives of grammar and content. Rephrase.
Line 70: What basal membrane is referred to?
Line 76: The listed factors are mostly proinflammatory which would be anti-tumor. Please clarify this discrepancy.
The immune cell section (2.2): M1 and M2 responses should be briefly explained. Furthermore, the principles immune checkpoint inhibition therapy should be described.
Section 2.3: A common feature of tumor vascular biology is leakage. This should be mentioned.
Fig 1: The color scheme makes CAFs hard to visualize.
Lines 160-162: The statement on MMPs is contradictory (both angiogenic and anti-angiogenic) and requires clarification.
Table 1: TGFb is pleiotropic and are the authors certain that its role in the context of the table is mainly inflammatory?
Line 328: What cells in the carcinomas respond favorably to ATP?
Line 372: Do the authors mean contradictory instead of controversial?
Lines 440-431, 531-532: The statement on chemoresistance should be modified. Although resistance may be less likely to develop, the cells may be more resistant to start out with due to a lower proliferative state.
Author Response
Reviewer 2
1. Is there information on stroma specifics in breast cancer subgroups, ie BRCA, ER+, PR+ HER2+, ductal, lobular etc?
Following the Referee recommendation, we have added in the section 3.1 page 5, lines 199-202 the following sentence and the corresponding references: independently of luminal or basal-like phenotype of breast carcinomas [91]. However, variations in MMP/TIMP expression among the different histological subtypes of breast carcinomas (ductal, lobular, mucinous, tubular, papillary and medullary invasive carcinomas) were found [92].”
2. MSC is mostly referred to as mesenchymal stem cells but actually in the vast majority of studies these cells have not been demonstrated as true stem cells. “Mesenchymal stromal cells” is more appropriate usage unless the cells in question have been demonstrated to be true stem cells or at least this distinction should be clarified. This applies to the text throughout the manuscript.
As recommended by the Referee, we have changed “stem cells” by “stromal cells” throughout the manuscript.
Minor points
Lines 24-25 and 48-49: “autonomous progression” and “cellular sociology” I find dubious from both the perspectives of grammar and content. Rephrase.
According to the Referee indication, we have changed the expression “autonomous progression” by “autonomous or acquired proliferative capacity” (line 24) and we have rephrased de sentence of line 48-52: “It seems increasingly clear that the old dogma of cancer based only on a malignant transformation of the epithelial cells is too simplistic, and a new concept considering cancer as an ecosystem based on a cell sociology and the tumor-stroma crosstalk, is gaining strength.”
Line 70: What basal membrane is referred to?
According to the Referee indication, we have corrected “basal membrane” by “basement membrane” and defined it as the interface of epithelium and stroma (page 2, lines 72-73).
Line 76: The listed factors are mostly proinflammatory which would be anti-tumor. Please clarify this discrepancy.
In the section 2.2 (Immune cells), we discuss the pro-tumoral effect of immune cells, therefore factors secreted by CAFs induce immune cell recruitment that may contribute to tumor progression, as indicated in the new version of the manuscript (page 2, lines 79-80).
The immune cell section (2.2): M1 and M2 responses should be briefly explained. Furthermore, the principles immune checkpoint inhibition therapy should be described.
Following the Referee recommendation we have briefly explained M1 and M2 responses (page 3, lines 96-101) : “Macrophages can be polarized into two phenotypes: classically activated (M1) macrophages and alternatively activated (M2) macrophages driven by cytokine repertoire of T helper cells (Th1 or Th2). It has been established M1 as a tumor-suppressive phenotype and M2 as a tumor-promoting phenotype, considering that TAMs are primarily M2 polarized.” Also, we have commented and added a reference regarding immune checkpoint inhibition (page 3, lines 115-118): “Due to the important role of host immune system in cancer, immune checkpoint inhibitors have garnered attention in the last years, specially against cytotoxic T-lymphocyte antigen-4 (CTLA-4) and programmed cell death protein 1 (PD-1) or its ligands (PDL-1).”
Section 2.3: A common feature of tumor vascular biology is leakage. This should be mentioned.
As recommended by the Reviewer, we have mentioned the vascular permeability and its consequences (page 3, lines 122-124): “Endothelial barrier maintains vascular and tissue homeostasis but its alteration lead to vascular permeability drives tumor-induced angiogenesis, blood flow disturbances, inflammatory cell infiltration, and tumor cell extravasation.”
Fig 1: The color scheme makes CAFs hard to visualize.
Following the Referee recommendation, we have changed the color of CAFs in the figure 1.
Lines 160-162: The statement on MMPs is contradictory (both angiogenic and anti-angiogenic) and requires clarification.
According to the Referee indication, we have tried to clarify the statement, but MMPs have both pro and anti-angiogenic capacity as show the new reference added. “MMPs also positively regulate cancer-related angiogenesis, through their ability to mobilize or activate proangiogenic factors (bFGF, VEGF, TGFβ and integrin αvβ3) after the degradation of basement membrane or ECM components and negatively via generation of angiogenesis inhibitors, such as angiostatin and endostatin.” (page 5, lines 182-183).
Table 1: TGFb is pleiotropic and are the authors certain that its role in the context of the table is mainly inflammatory?
According to the Referee comment, we have added in the new table 2 the role of TGFb in tumor invasion.
Line 328: What cells in the carcinomas respond favorably to ATP?
Following the Referee recommendation, we have indicated increase in extracellular ATP levels “(…) induces a favorable effect in migration and invasion of breast cancer cells.” (page 11, line 372).
Line 372: Do the authors mean contradictory instead of controversial?
We have changed “controversial” by “contradictory” (page 12, line 415).
Lines 440-431, 531-532: The statement on chemoresistance should be modified. Although resistance may be less likely to develop, the cells may be more resistant to start out with due to a lower proliferative state.
We have changed the expression by “long-term chemoresistance” (lines 484 and 575).
Reviewer 3 Report
In the submitted paper entitled “Breast cancer tumor stroma: cellular components, phenotypic heterogeneity, intercellular communication, prognostic implications and therapeutic opportunities” the authors are showing new approaches to better understand the prognosis of the cancer disease.. As point of interest the authors prose new results and evidences to support the theory of cancer as an ecosystem based on a cellular sociology, with diverse components and complex interactions between different stroma cell types.
The authors point out the relevant role in the dynamics of tumor/stromal cell interactions, fibroblasts, endothelial cells, immune cells and mesenchymal stromal/stem cells and they focus the metastasis development due to matrix metalloproteases and their regulators. The studied bi-directional signals between cancer cells and tumor stroma cells may suggest new therapeutic strategies.
The paper is very well and clearly written, easy to read, of valuable interest and useful for the readers of Cancer and it is able to summarize the best evidences in this topic proposing a two types pathophysiological model of stroma breast carcinoma, further the figures are very clear and they are able to represent the main process and route reported.
Still of particular interest the role played by MSC derived exosomes in the comprehension of the full picture of paracrine communications, the sections 4.2, 4.3 and 4.4 in the chapter 4 should be summarized and shorted. Further the section 3.1, 3.2, 3.3 in the chapter 3 may be summarized in a one section.
Author Response
Reviewer 3
Still of particular interest the role played by MSC derived exosomes in the comprehension of the full picture of paracrine communications, the sections 4.2, 4.3 and 4.4 in the chapter 4 should be summarized and shorted. Further the section 3.1, 3.2, 3.3 in the chapter 3 may be summarized in a one section.
Sorry, but we cannot respond correctly to these suggestions because the Editor has “appreciated the information especially on the exosomes” and he suggested to develop the information about MMPs, so we can not summarize or shorten these sections.